# Superparamagnetic Fe_3_O_4_@CA Nanoparticles and Their Potential as Draw Solution Agents in Forward Osmosis

**DOI:** 10.3390/nano11112965

**Published:** 2021-11-04

**Authors:** Irena Petrinic, Janja Stergar, Hermina Bukšek, Miha Drofenik, Sašo Gyergyek, Claus Hélix-Nielsen, Irena Ban

**Affiliations:** 1Faculty of Chemistry and Chemical Engineering, University of Maribor, Smetanova 17, SI-2000 Maribor, Slovenia; janja.stergar@um.si (J.S.); hermina.buksek@um.si (H.B.); miha.drofenik@um.si (M.D.); saso.gyergyek@um.si (S.G.); irena.ban@um.si (I.B.); 2Institute of Biomedical Sciences, Faculty of Medicine, University of Maribor, Taborska Ulica 8, SI-2000 Maribor, Slovenia; 3Department of Materials Synthesis, Jožef Stefan Institute, Jamova cesta 29, SI-1000 Ljubljana, Slovenia; 4Department of Environmental Engineering, Technical University of Denmark, Miljøvej 113, 2800 Kgs. Lyngby, Denmark

**Keywords:** citrate-coated magnetic nanoparticle, forward osmosis, draw solution, osmotic pressure, non-ideality analysis

## Abstract

In this study, citric acid (CA)-coated magnetite Fe_3_O_4_ magnetic nanoparticles (Fe_3_O_4_@CA MNPs) for use as draw solution (DS) agents in forward osmosis (FO) were synthesized by co-precipitation and characterized by Fourier transform infrared spectroscopy (FTIR), thermogravimetric analysis (TGA), dynamic light scattering (DLS), transmission electron microscopy (TEM) and magnetic measurements. Prepared 3.7% *w*/*w* colloidal solutions of Fe_3_O_4_@CA MNPs exhibited an osmotic pressure of 18.7 bar after purification without aggregation and a sufficient magnetization of 44 emu/g to allow DS regeneration by an external magnetic field. Fe_3_O_4_@CA suspensions were used as DS in FO cross-flow filtration with deionized (DI) water as FS and with the active layer of the FO membrane facing the FS and NaCl as a reference DS. The same transmembrane bulk osmotic pressure resulted in different water fluxes for NaCl and MNPs, respectively. Thus the initial water flux with Fe_3_O_4_@CA was 9.2 LMH whereas for 0.45 M NaCl as DS it was 14.1 LMH. The reverse solute flux was 0.08 GMH for Fe_3_O_4_@CA and 2.5 GMH for NaCl. These differences are ascribed to a more pronounced internal dilutive concentration polarization with Fe_3_O_4_@CA as DS compared to NaCl as DS. This research demonstrated that the proposed Fe_3_O_4_@CA can be used as a potential low reverse solute flux DS for FO processes.

## 1. Introduction

Magnetic nanoparticles (MNPs) have attracted attention in research and industry in the chemical, environmental and medical fields. MNPs have shown promising performance in removing contaminants or reducing toxicity [1,2,3,4,5] and have generated interest in research into technical applications for treatment of polluted water and water purification processes [6,7,8,9,10]. In recent years, MNP’s have also attracted attention as materials for generating the driving force to transport water in a forward osmosis (FO) membrane separation process, one of the emerging membrane technologies that can meet the increasing global demand for water recycling and reuse [11]. FO is a technical term describing a natural phenomenon of osmosis: the transport of water molecules across a semi-permeable membrane [12]. In contrast to pressure-driven membrane processes, FO is driven by the osmotic pressure difference across the FO membrane. In ideal FO, a concentrated draw solution (DS) with high osmotic pressure extracts water from a dilute feed solution (FS) while ideally rejecting all FS solutes [13,14]. In order to function effectively as a draw agent in FO, the osmotic pressure of the DS must significantly exceed that of the FS. Simple inorganic salts, such as NaCl, remain the most widely used DS agents due to their ability to have high osmotic pressures while maintaining low solution viscosities. The strong affinity of small inorganic ions for water is reflected in their highly exothermic enthalpies of hydration [15]. Strong solvent-solute interactions provide high solution osmotic pressures while making the regeneration of DS more difficult. In resolving this problem, the development of easily removable DS agents, which allow for regeneration through exploitation of solute size, thermal sensitivity or magnetic properties, is desirable [16].

Although the FO technology is gaining popularity in niche applications including cold-concentration in the food and beverage industry, drawbacks, such as high reverse solute flux and high regeneration costs, still restrict wider application [17,18,19]. A reverse solute flux can potentially contaminate the FS apart from decreasing the osmotic pressure difference across the membrane. The suitability of a draw solute is defined by the ability to develop sufficient osmotic pressure in order to ensure a high water flux while maintaining low (ideally no) diffusion from the DS to the FS. The molecular size and ionic structure of the DS agent define its applicability in FO. A smaller molecular size of the DS agents reduces the internal concentration polarization and thus associated with a higher water flux however, this is generally also associated with higher reverse solute flux. In comparison, larger DS agent molecules are associated with less internal concentration polarization (and thus lower water flux) but also a lower reverse solute diffusion [20], the latter a prerequisite for use in cold-concentration.

Over the past years, a variety of compounds have been investigated as draw solutes, including sulfur dioxide [21], various inorganic salts and a variety of sugars [22,23], thermally unstable ammonium salts [24,25] polyelectrolytes [26] and MNP’s [27,28,29]. Iron oxide MNPs such as maghemite (γ-Fe_2_O_3_) and magnetite (Fe_3_O_4_) MNPs have been widely studied due to well-defined properties such as small uniform particle size in the range of 10 to 100 nm, biocompatibility, heat generation in response to an alternating magnetic field (AMF), easy surface manipulation for targeted release, low toxicity profile, and good colloidal stability [30]. Although pristine iron oxide MNPs have desirable properties (size and stability) as DS they cannot generate significant osmotic pressure without appropriate surface modification [31,32]. To improve water solubility and surface hydrophilicity, MNP’s functionalized with strong hydrophilic groups are considered as one of the feasible solutions to generate sufficient osmotic pressures, as well as allow for facile regeneration [33] as exemplified by Fe_3_O_4_ MNPs, which can be easily separated from water by an external magnetic field [34]. To maximize the osmotic pressure that is generated from the functional groups coated on the iron oxide MNPs, a high number of polymer hydrophilic groups must be available to interact with the solvent water molecules. Citric acid (CA) is a small molecule which is suitable for facile functionalization and confers hydrophilic properties to MNPs by virtue of three –COOH groups with one or two –COOH groups absorbed on the surface of the MNPs and at least one remaining free [32,35,36,37]. Recently Khazaie et al. and his co-workers synthesized Fe_3_O_4_ nanoparticles and covalently functionalized them with tri-sodium citrate. The DS with an optimal ratio of CA: MNPs (2:1) with concentrations of 20, 40, 60, and 80 g L^−1^ showed water fluxes of 7.1, 10.7, 14.2, and 17.1 LMH, with osmotic pressures of 114, 117, 120 and 124 bar respectively [38]. Applications with MNP coated with sodium citrate has also reported by [39,40,41].

Here we investigate the synthesis of Fe_3_O_4_ MNP’s coated with CA and evaluate their feasibility as DS agents in FO. MNP’s were synthesized by co-precipitation of Fe^2+^ and Fe^3+^ aqueous solutions with hydrophilic CA by adding a base in a one-pot synthesis. The research is comprised of two sections: (1): the synthesis of Fe_3_O_4_@CA MNP’s with characterization using FTIR, TGA, DLS, TEM and magnetic measurements and (2): DS testing using an AIM™ hollow fibre FO (HFFO) module where water flux and reverse solute flux is benchmarked against NaCl as DS using deionized (DI) water as the FS.

## 2. Materials and Methods

### 2.1. Materials

The chemicals used for the synthesis of Fe_3_O_4_ MNP’s coated with CA were iron (III) chloride hexahydrate (FeCl_3_ × 6H_2_O, Sigma Aldrich, Darmstadt, Germany), iron (II) chloride tetrahydrate (FeCl_2_ × 4H_2_O, Sigma Aldrich), sodium hydroxide (NaOH, Sigma Aldrich) and CA (C_6_H_8_O_7_, Laiwu Taihe Biochemistry Co., Laiwu, Shadong Province, China). All the chemicals used in the experiments were of analytical grade. Inert nitrogen (N_2_, Messer, Ruše, Slovenia) atmosphere was used during synthesis. Ethanol (C_2_H_5_OH, Sigma Aldrich) was used as washing materials. DI water is used throughout all experiments. The chemical used for FO filtrations was sodium chloride (NaCl, LabExpert, Berdyansk, UK). DI water was used for NaCl solution preparation, for cleaning the membrane module, and as FS during performing baseline measurement.

### 2.2. Synthesis of Fe_3_O_4_@CA MNPs

For the synthesis of Fe_3_O_4_@CA MNPs a typical co-precipitation process was applied [35]. A 25 mL solution containing 1.28 M Fe^3+^, 0.64 M Fe^2+^ (molar ratio 2:1), and 1.28 M CA was placed in a three-necked flask at 80 °C (oil bath) with vigorous stirring (340 rpm) for 20 min under N_2_ atmosphere. Then 250 mL 1 M NaOH solution was added drop by drop using a dropping funnel and the solution was continuously stirred for an additional 60 min at 80 °C under an N_2_ atmosphere. The reaction scheme is as follows (Equation (1)):(1)$${\mathrm{FeCl}}_{2}\;+\;2{\;\mathrm{FeCl}}_{3}\;+\;8\;\mathrm{NaOH}\;\to {\mathrm{Fe}}_{3}O_{4}↓\;+\;8{\;\mathrm{Na}}^{+}\;+\;8{\;\mathrm{Cl}}^{-}\;+\;4{\;H}_{2}O$$

In the final product, in addition to the desired Fe_3_O_4_ MNP’s, a large amount of dissolved sodium and chloride ions is also present. To reduce their concentration, the suspension was washed several times as described here (discussed in more details in Section 3.1). Fe_3_O_4_@CA MNPs were uniformly dispersed in 100 mL of DI as a stock solution by 30-min sonication in an ultrasonic bath. The synthesis process is shown schematically in Figure 1.

### 2.3. Characterization of Fe_3_O_4_@CA MNP’s

The functional groups of Fe_3_O_4_@CA MNP’s were analysed by Fourier transform infrared spectroscopy (FTIR, mod. 5000, Perkin-Elmer Inc., Beaconsfield, UK). The FTIR spectrum was measured in the range 4000–400 cm^−1^. The weight loss of Fe_3_O_4_@CA MNP’s was characterized by thermogravimetric analysis (TGA, TGA/SDTA, 851^e^ Mettler Toledo, Greifensee, Switzerland). The measurement was performed under air from 30 to 800 °C at a heating rate of 10 K/min. The average hydrodynamic particle size, zeta potential, and isoelectric point (IEP) were determined by dynamic light scattering (DLS, Zetasizer Nano ZS, Malvern, Worcestershire, UK) using purified Fe_3_O_4_@CA MNP’s water dispersions at 0.1 g/L with DI and pH adjusted from the current pH value to 2.0 during the measurements with an auto titrator [42]. The size and morphology of the nanoparticles were investigated by transmission electron microscopy (TEM, JEM-2010F, JEOL, Tokyo, Japan). The Fe_3_O_4_@CA MNP suspension was deposited on a copper-grid-supported carbon film specimen holder and left to dry at ambient conditions. The empirical size distribution of the MNPs was estimated by measuring an area of the MNP’s on the TEM image. The average particle size *d*_TEM_ is given as a number-weighted average equivalent diameter (from Gaussian fit of the empirical distribution), the diameter of a circle having the same area as the imaged particle. The surface ligand concentration *n_s_* (the number of ligands per particle) were estimated using Equation (2) [43]:(2)$$n_{s}\;=\;\omega \times 1/6\pi d^{3}\rho_{Fe_{3}O_{4}}\;\times \;6.023/M_{CA}\left(1-\frac{\omega }{100}\right)$$
where, *ω* is weight loss (%), *d* is particle diameter (nm), estimated from the TEM images to be 3–7 nm assuming spherical-shaped nanoparticles, *ρ**_Fe3O4_* is 5.18 g/cm^3^ and MCA is 192.14 g/mol.

The room temperature magnetization curve of the Fe_3_O_4_@CA MNP’s as a dry powder was measured using a vibrating sample magnetometer (VSM; model 7307, Lake Shore Cryotronics, Westerville, OH, USA). The saturation magnetization Ms of the sample is given as the average of the magnetiza-tions measured at magnetic field strength H of −10 kOe and 10 kOe.

The osmolality of the DS prepared from MNP’s was measured with an osmometer (Gonotec-Osmomat 030, Berlin, Germany) and the osmotic pressure, π of Fe_3_O_4_@CA MNP’s solutions were calculated using Equation (3):(3)π = (∑φ × n × c) × R × T
where *R* is gas constant (8.31447 J/mol K), *T* is the absolute temperature (K), *n* is the amount of substance (mol), *c* is the molarity of a solution (mol/L), and ϕ is an osmotic coefficient (-). The characterisation methods in detailed is presented in a previous paper [29].

### 2.4. FO Filtrations

The FO filtration experiments were performed using the AIM™ HFFO module. The specification of the AIM™ HFFO module and experimental set-up used in the study was presented in a previous paper [44]. The cross-flow velocity used for experiments was 120 mL/min in co-current mode. Characteristics of used DS and FS are presented in Table 1.

Water flux, *J_w_* (LMH) across the membrane was calculated using Equation (4) [45]:(4)$$J_{w}\;=\;\frac{\Delta V}{A\;\Delta t}$$
where: Δ*V* total volume change of permeate water (L), *A* effective membrane area (m^2^) and Δ*t* is the time (h).

The reverse salt flux, *J_s_* (GMH) was determined using Equation (5) [45]:(5)$$J_{s}\;=\;\frac{\gamma_{t}V_{t}-\gamma_{0}V_{0}}{A\;\Delta t}$$
where: *γ*_0_ = initial concentration of the FS (g/L), *V*_0_ = initial volume of the FS (L), *γ_t_* = solute concentration at time *t* (g/L), *V_t_* = volume of the FS measured at time *t* (L), *A* = effective membrane area (m^2^) and Δ*t* = time (h).

The recovery fraction (*R*) indicates the amount of feed recovered as permeate using Equation (6) [46]:(6)$$R\;\left(\%\right)\;=\;\frac{V_{p}}{Vf}\;100$$
where: *V_P_* = the permeate volume of water (L), and *V_F_* = the volume of water in the FS (L).

## 3. Results and Discussion

The results are presented as follows: First, MNP’s were synthesized by co-precipitation of Fe^2+^ and Fe^3+^ aqueous solutions with hydrophilic CA by adding a base in a one-pot synthesis, see Figure 1. Second, the as-prepared Fe_3_O_4_@CA MNP’s were fully characterized using TEM, FTIR, TGA, DLS, zeta potential and magnetic measurements (Section 3.1). Third, the non-ideality of the solutions of un-grafted and grafted CA molecules onto MNP’s was investigated (Section 3.2) and the osmotic properties and FO performance described (Section 3.3). Finally, FTIR and TGA analyses were done of Fe_3_O_4_@CA MNP’s after the FO process (Section 3.4).

### 3.1. Magnetic Particle Preparation

After synthesis, the Fe_3_O_4_@CA MNPs solution was cooled to room temperature (21 °C), and the black product was separated by a permanent magnet (0.8 T). Fe_3_O_4_@CA MNPs were rinsed out four times with DI water, once with ethanol and finally with DI again. In Table 2 conductivity, pH and osmotic pressures of Fe_3_O_4_@CA MNPs after synthesis and washing cycles are presented.

From Table 2, the changes in conductivity, pH and osmotic pressure after synthesis and multiple washing with ethanol and water can be followed (explained in more details in the Results (Section 3.4.2.)).

### 3.2. Characterization of Fe_3_O_4_@CA MNPs

The coordination of the CA (carboxylic groups) to the nanoparticle surface is confirmed by FTIR spectroscopy (Figure 2). In the FTIR spectrum of bare MNPs (black), the band at 580 cm^−1^ corresponds to the vibration of the Fe-O bonds in the crystal lattice of Fe_3_O_4_ (magnetite) [47,48]. A broad band at 3400 cm^−1^ can be assigned to the structural OH groups of water molecules [36]. The FTIR spectrum of pure CA (red) dispersive and broadened O-H stretching vibration at 3400 cm^−1^ and C=O stretching vibration of (R-COOH) group at 1739 cm^−1^ are present. Asymmetrical and symmetrical stretching vibrations of the carboxylate group were observed at 1547 cm^−1^ and 1386 cm^−1^, respectively.

Compared to bare MNPs, two additional peaks were observed for Fe_3_O_4_@CA MNP’s at the wavelengths 1600 cm^−1^ and 1400 cm^−1^ (blue), representing the symmetrical and asymmetrical stretching of C=O and CO vibration of the three carboxyl groups (-COOH) as well as the large and intense band from 3200 to 3400 cm^−1^ corresponding to the OH group confirms the presence of non-dissociated OH groups of the CA and water traces [36,49,50]. Thus, our FTIR measurements confirmed that the CA binds chemically to the magnetite surface by carboxylate chemisorption and citrate ions are formed.

A thermogravimetric analysis (TGA) was performed to measure the organic content of Fe_3_O_4_@CA MNP’s. The results of the TGA are shown in Figure 3 which shows the weight loss of Fe_3_O_4_@CA. Here, the correction (3%) due to the oxidation of magnetite in hematite was taking into the account and the mass loss of the acetic acid was therefore accordingly corrected.

The weight loss curve observed in a range of 100 to 800 °C is attributed to the evaporation of water, the decomposition of the bound citrate molecules on the Fe_3_O_4_ MNPs’ surfaces 3% [41,51] and to oxygen during magnetite oxidation 3.3%. The initial weight loss below 160 °C due to the evaporation of physically adsorbed water was 5% and the organic content of coated Fe_3_O_4_@CA was determined to be 6.3%. This is the mass percentage of all surface coated organic groups on the MNPs corresponding to desorption of CA molecules on the magnetite particles. The number of COOH groups bound to the surface of the nanoparticles can be calculated according to Equation (2). Assuming a spherical shape of the MNPs with an average diameter of five nm, then the estimated total number of COOH groups is 71 per particle.

The surface charges of the bare Fe_3_O_4_ MNPs and Fe_3_O_4_@CA MNPs were characterized by zeta potential measurements as a function of pH from 7 to 2 and z-average hydrodynamic diameter changes as a function of pH. Figure 4 presents the results for Fe_3_O_4_@CA MNPs.

As reported in the literature, the isoelectric point (IEP) for neat Fe_3_O_4_ is ~6.8 [41,52]. Fe_3_O_4_@CA MNP’s have a negative zeta potential of −30 mV at pH = 7 as shown in Figure 4. This observed phenomenon is likely caused by the adsorption of CA molecules on the bare MNPs’ surface, where the surface charges were influenced by the introduction of carboxylate groups [41,53]. The zeta potential value becomes more negative with increasing pH due to the increase of OH^–^ ions in the dissociated solution and deprotonation of the carboxyl groups of CA. This confirmed the presence of negatively charged carboxylate groups’ on the MNPs surface. The ensuing electrostatic repulsion ensures their colloidal stability in aqueous suspension where some of the carboxylate groups from CA are adsorbed/coordinated on the surface of MNPs while uncoordinated species protrude into the water medium [54]. Thus, the effective charge at pH 7 provides stabilization of Fe_3_O_4_@CA MNPs. The effective hydrodynamic diameter of Fe_3_O_4_@ CA MNPs, measured by DLS at 25 °C, changes from 500 to 3500 nm with varying pH. Since the pH was adjusted with HCl, the effective particle size increased when the IEP 4.93 was reached and then decreased back to the original size when the pH dropped below the IEP 4.93.

The osmotic pressure of the Fe_3_O_4_@CA solution was determined by the freezing point depression method. Table 3 shows concentration, osmotic pressure, size, number of molecules (*n_s_*), IEP, and magnetic saturation of Fe_3_O_4_@CA MNPs.

Morphology and size of the nanoparticles Fe_3_O_4_@CA were observed using TEM. Upon drying on the TEM specimen support the nanoparticles formed agglomerates (Figure 5a,b). However, at the edges of the agglomerates, the deposit of nanoparticles is relatively thin and individual nanoparticles can be resolved (Figure 5b). The nanoparticles are approximately spherical and their estimated average diameter *d*_TEM_ is 5.2 nm ± 0.9 nm (Figure 5c).

The room-temperature magnetization curve of the nanoparticles Fe_3_O_4_@CA suggests a superparamagnetic state of the nanoparticles, due to the lack of measured coercivity and remanence (Figure 6). The saturation magnetization of the Fe_3_O_4_@CA *M*_s_ of 45 emu/g is significantly smaller than reported values for bulk Fe_3_O_4_, which are in the range between 92 emu/g and 100 emu/g [54]. In the Fe_3_O_4_@CA magnetic nanoparticles are “diluted” with diamagnetic physically adsorbed water and CA (Figure 4) which reduces the measured *M*_s_. The TGA analysis suggests that the diamagnetic mater contributes approx. 8 wt.% of the simple Fe_3_O_4_@CA, therefore estimated value of *M*_s_ of the Fe_3_O_4_ is 49 emu/g. The value is still smaller than value characteristic for bulk material and is consistent with small size of the Fe_3_O_4_ MNPs [55]. Because of the surface effects, namely the magnetically distorted surface layer small nanoparticles in general exhibit lower values of saturation magnetization than bulk material of the same composition [56].

### 3.3. Properties of Fe_3_O_4_@CA

The advantage of coated nanoparticles is that they may have higher osmotic pressures than solutions consisting of the grafting agents alone at the same concentration [16]. The increase in osmotic pressure values can be attributed to the increased solvent-accessible surface area and thus improved hydration. The non-ideality parameters *I* and *S* were determined for Fe_3_O_4_@CA, see Figure 7 and *M**_CA_ calculated using the semi-empirical model Equations (7) and (8):(7)$$\frac{m_{w}}{m_{s}}\;=\;S\;\times \;\frac{1}{\pi }\;+\;I$$
(8)$$S\;=\;\frac{R\;\times \;T\;\times \;\rho }{M^{*}}$$
where *m_w_* and *m_s_* is the mass of water and solute respectively; *S* and *I* are the non-ideality parameters, π the osmotic pressure, ρ is the density of water at temperature *T*, *M* is the molecular weight of the solute and *R*, *T* gas constant and temperature.

The results show that the osmotically derived molecular weights are *M**_CA_ = 176.12 g/mol for pure CA and *M**_CA_ = 57.9 g/mol for MNP@CA. The value *M**_CA_ = 57.9 g/mol is significantly lower than the nominal standard value of 192.12 g/mol, indicating that the MNP-grafted CA molecule are almost four-fold stronger as osmotic agent as CA alone.

### 3.4. Forward Osmosis Process Evaluation

#### 3.4.1. Determination of *J_w_* and *J_s_*

Fe_3_O_4_ MNPs at concentrations of CA of 3.7% were used to prepare Fe_3_O_4_@CA as DS in a lab-scale cross-flow FO filtration setup. In Figure 8 water flux is presented where DI was used as FS and Fe_3_O_4_@CA as well as the 0.45 M NaCl was used as DS, respectively, in order to make a comparison between Fe_3_O_4_@CA solution and standard DS (NaCl solution). For this purpose, we wanted to imitate the FO process when using Fe_3_O_4_@CA solution as DS, including measuring conditions (e.g., starting volumes of FS and DS, initial osmotic power of DS). Imitation was performed using NaCl solution as DS, therefore, 0.45 M concentration of NaCl was used to give initial osmotic pressure comparable to the solution of Fe_3_O_4_@CA. The filtrations with Fe_3_O_4_@CA were stopped after 2 h to ensure sufficient data to have an insight into the filtration dynamics. The filtration with 0.45 M NaCl as DS was finished spontaneously in 1 h (75.2 g of FS left) because the water flux was higher in comparison when Fe_3_O_4_@CA was used as DS agent despite the similar initial osmotic transmembrane gradient.

The initial water flux of 0.45 M NaCl solution as reference is higher (14.1 LMH) when comparing to the Fe_3_O_4_@CA (9.3 LMH). A uniform water flux decline was observed for all solutions as a function of time. The overall water flux decreased 30% and 70% for 0.45 M NaCl and Fe_3_O_4_@CA, respectively, due to dilution of DS and reduction in the osmotic pressure difference between FS and DS.

When Fe_3_O_4_@CA was used as DS, the diffusion of water molecules decreased due to the higher concentration polarization of the Fe_3_O_4_@CA from the draw side (96.1 mL), presented in Table 5. Therefore, the dilutive ICP in FO mode is stronger for Fe_3_O_4_@CA and the effective osmotic gradient across the active layer is lower, see Figure 9.

The decrease in water flux with an increase in the conductivity indicates that the main driving force across the FO membrane is the osmotic pressure difference between the DS and FS. The conductivity was monitored (Figure 10a) and reverse solute flux, *J_s_* (Figure 10b) was calculated using Equation (5) where concentrations in FS were obtained out of measured conductivity data in the FS. Although the 0.45 M NaCl solution gives rise to higher water flux (29.1% higher) compared to Fe_3_O_4_@CA, the reverse solute flux is also larger (2.5 versus 0.08 GMH). The conductivity increased (as well as *J_s_*) for the 0.45 M NaCl solution during one hour.

The suitability of a draw solute is defined by its ability to develop osmotic pressure, which leads to higher water flux, and lower diffusion to the feed side. From the molecular view, the size and ionic structure of the draw solute defines its applicability in the FO process. The lower molecular size increases the diffusivity of the DS and reduces the internal concentration polarization that leads to the higher water flux, while the reverse salt flux is also enhanced. The larger molecules are designed as DS to prevent reverse salt diffusion.

In this study, small molecules such as NaCl that were used for comparison generated high osmotic pressure at low solution viscosities and mitigate ICP due to their high reverse solute flux [20].

The synthesized Fe_3_O_4_@CA particles are relatively large, and the reverse diffusion was low, however, the potency of DS to build up a reasonable osmotic pressure and water flux was enhanced by increasing particle hydrophilicity.

The used AIM™ HFFO module was cleaned using only DI water for 30 min and then a new baseline experiment was performed under the same conditions. The water flux of the cleaned AIM™ HFFO module (restored to about 97%) was almost the same as that of the pristine module. This indicates that simple physical flushing can effectively rejuvenate the used module membrane and the average water flux can be restored to about 93.2% even after 2 cycles.

#### 3.4.2. Development of the FS and DS Osmotic Pressures

The osmotic pressure of the Fe_3_O_4_ functionalized with CA is relatively high due to the higher hydrophilicity because the CA molecule possesses two terminal COOHs, one central COOH, and one C−OH group, which are all active in bonding with iron oxide via the Fe−OH groups [35,36]. For a given concentration and temperature, the osmotic pressure depends on the solubility and molecular weight of the DS moieties. From the CA (H_3_C_6_H_5_O_7_) speciation, CA will be mostly deprotonated as HC_6_H_5_O_7_^2−^ and C_6_H_5_O_7_^3−^ over a broad pH range (5.0–9.0). Thus, at pH = 7 there will be mostly HC_6_H_5_O_7_^2−^ and C_6_H_5_O_7_^3−^ functional groups of CA [57]. The deprotonated CA made MNP@CA surfaces highly negatively charged [32]. However, both MNP@CA (and CA alone) will have counter ions closely attached and will thus not contribute to bulk solution conductivity as such.

When comparing the development in FS conductivity (Figure 10a) with the FS osmotic pressure development (Figure 11a) it is observed that the osmotic pressure of the FS with 0.45 M NaCl as DS reached 0.5 bar in one hour while at the same time, the osmotic pressure for the FS with Fe_3_O_4_@CA as DS reached 4 bar, respectively. This suggests that DS solutes diffuse across the membranes into the FS. The DS contains four components: Fe_3_O_4_@CA MNP, unconjugated CA, ethanol and NaOH. As the conductance increase in the FS is negligible (compared to NaCl) the reverse solute flux could in principle be due to any of the three first components or a combination thereof (since NaOH would be dissociated and thus give rise to an increase in solution conductivity). However, a 4 bar FS osmotic pressure at the end of the experiment (see Figure 12a) generated solely by Fe_3_O_4_@CA MNPs would correspond to a 0.8% *w*/*w* solution which is opaque. Since the FS is transparent it is unlikely that Fe_3_O_4_@CA MNP’s are crossing the membrane. This leaves only ethanol and/or unconjugated CA as potential reverse flux solute candidates. The FS ethanol concentration was measured to be 0.07% which corresponds to an osmotic pressure of 2.8 bar. Thus, ethanol diffusion can explain some of the FS osmotic pressure increase during the experiment. With regards to unconjugated CA is crossing the membrane, it is noted that CA has a molecular weight of 192 Da which is relatively large as compared to ionic salts. However, it has previously been shown that peptides with molecular weights of 375 and 692 Da can cross a thin film composite Aquaporin Inside™ Membrane (AIM) FO membrane and the transport mechanism was determined to be diffusion-based [58]. Thus, it is likely that unconjugated CA (with Na^+^ as associated counter ion) also contributes to the increase in FS osmotic pressure.

When comparing with other studies where magnetic nanoparticles were used a DS in FO, several studies have focused on poly-sodium acrylate (PSA) coated MNPs as DS. Thus, a water flux of 5.3 LMH and a DS osmotic pressure of 11.4 bar has been reported using PSA-MNPs at 0.078%, wt% (~1.3 g/L) DS concentration ([57]. Ge and co-workers [59] reported an osmotic pressure of 11–12 bar at a concentration of ~15%, wt% PSA (Mn 1800) as DS. Similar osmotic pressures were obtained with 24–48%, wt% PSA DS (in the form of free polyelectrolyte osmotic agent) [31]. The significant difference in osmotic pressure between PSA-MNP solutions and free PSA solutions reflect the non-ideality of PSA-MNP solutions similar to what we observe in this study. However, the robustness of PSA-MNP based DSs (i.e., their ability to maintain their osmotic pressure after repeated concentrations) is still an issue [29].

In our previous study [29] a 7% solution of robust PSA-coated MNPs yielded an initial *J_w_* of 4.2 LMH and an osmotic pressure of 9 bar and a reverse solute flux *J_s_* of 0.05 GMH. Thus, the results presented here where Fe_3_O_4_@CA MNPs a 3.7% solution yielded approximately a two-fold higher initial *J_w_* (Figure 8) and osmotic pressure (Figure 9) implies that the Fe_3_O_4_@CA MNPs DS is about four-fold more potent as DS compared to a DS based on PSA-coated MNPs. However, the nominal reverse solute flux for Fe_3_O_4_@CA of 0.08 GMH (Figure 10b) is slightly higher than for PSA-coated MNP based DS.

### 3.5. Characterization after FO Process

The functional groups of chemical bonds between Fe_3_O_4_ and CA after the FO process were characterized by FTIR (Figure 13a). It is confirmed that the Fe_3_O_4_@CA spectral features are preserved, so the CA coating on Fe_3_O_4_ MNPs was maintained. The TGA results in Figure 13b show the weight loss of Fe_3_O_4_@CA before (black curve) and after the FO process (red curve). The overall weight loss of Fe_3_O_4_@CA before the FO process was determined 8.3% and after the FO process 6.6%. The total estimated number of COOH groups is still 85 molecules per particle after FO process.

## 4. Conclusions

The CA-coated Fe_3_O_4_ nanoparticles (Fe_3_O_4_@CA MNPs) have been synthesized for use as draw solution (DS) agents in forward osmosis (FO) and systematically analyzed. Non-ideality of the solutions of ungrafted and grafted CA molecules onto MNPs indicates that the MNP-grafted CA molecule are almost four-fold stronger as osmotic agent as CA alone. Carboxyl groups of CA have an affinity for MNPs, thus preventing aggregation. The osmotic pressure of as-prepared MNP@CA was high enough to be used as a DS in FO. Although the 0.45 M NaCl solution gives rise to higher water flux (for 29.1%) compared to Fe_3_O_4_@CA, the reverse solute flux is also larger (2.5 to 1.8 GMH). These differences are ascribed to a more pronounced internal concentration polarization associated with Fe_3_O_4_@CA as DS as compared to NaCl as DS. The results showed that the conductivity of the feed solution stays the same low level before and after FO filtration, and that the reverse solute flux of Fe_3_O_4_@CA MNPs is negligible. Still the FS osmotic pressure is increasing during the filtration, which is ascribed to diffusion of DS ethanol and unconjugated CA. Thus, an effective purification of Fe_3_O_4_@CA MNPs is a prerequisite in order to be able to benefit fully from the low inherent reverse solute flux. Therefore, FO can be considered as a potential candidate for a broad range of concentration applications where current technologies still suffer from critical limitations

## Figures and Tables

**Figure 1 nanomaterials-11-02965-f001:**
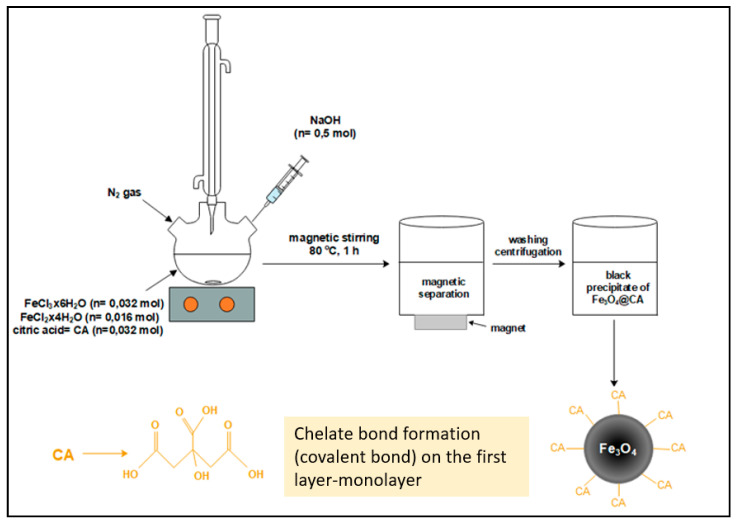
Schematic diagram of co-precipitation method of Fe_3_O_4_@CA MNP’s.

**Figure 2 nanomaterials-11-02965-f002:**
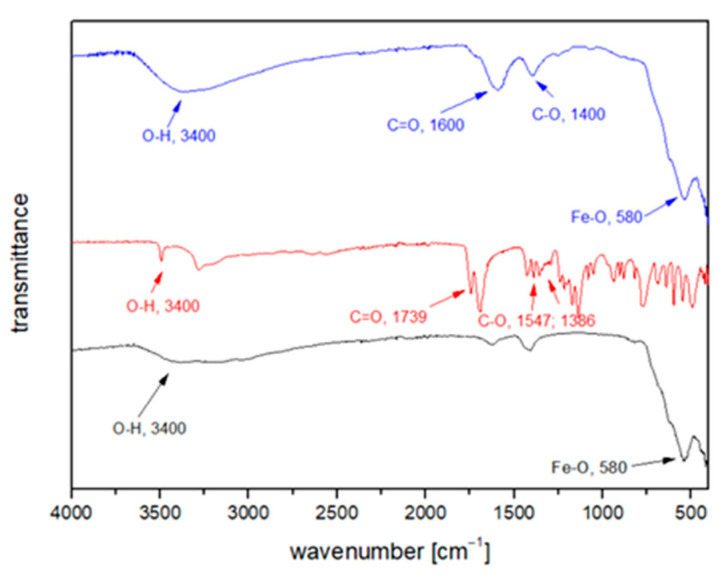
FTIR spectra of magnetite MNPs (black), CA (red), and Fe_3_O_4_@CA MNP’s (blue).

**Figure 3 nanomaterials-11-02965-f003:**
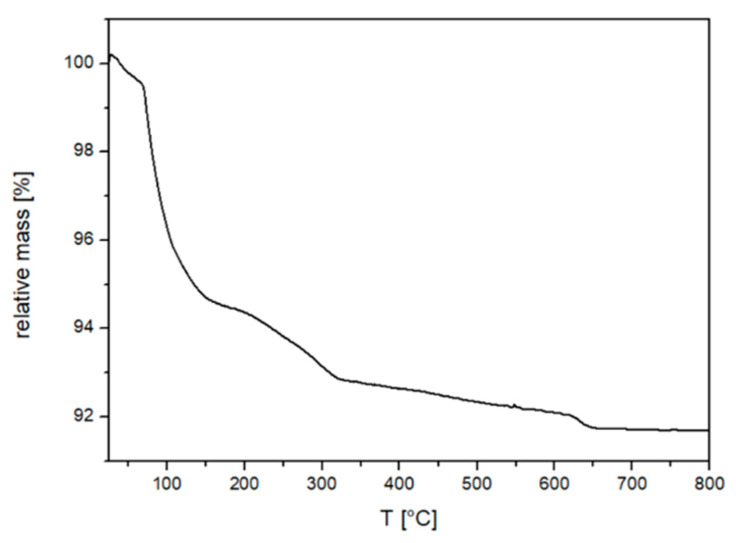
TGA curve of Fe_3_O_4_@CA MNPs.

**Figure 4 nanomaterials-11-02965-f004:**
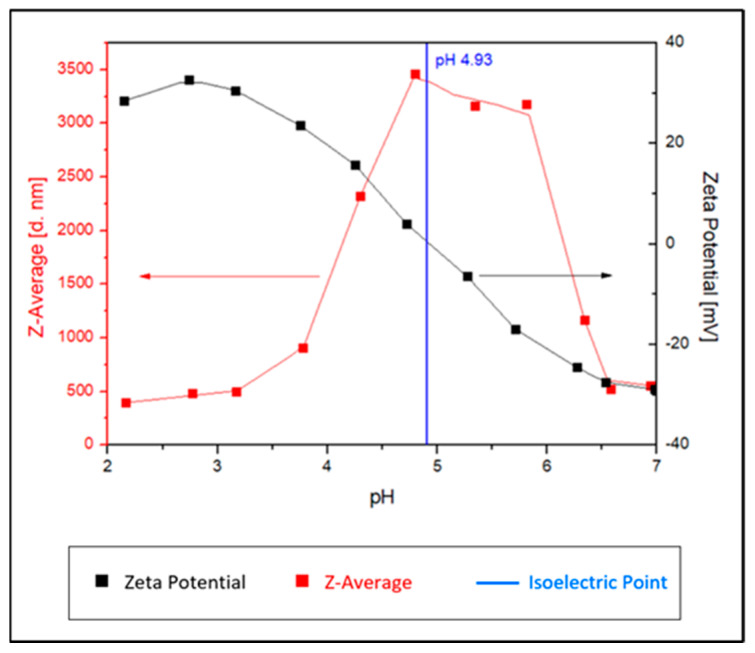
Change of zeta potential for Fe_3_O_4_@CA MNPs with pH values.

**Figure 5 nanomaterials-11-02965-f005:**
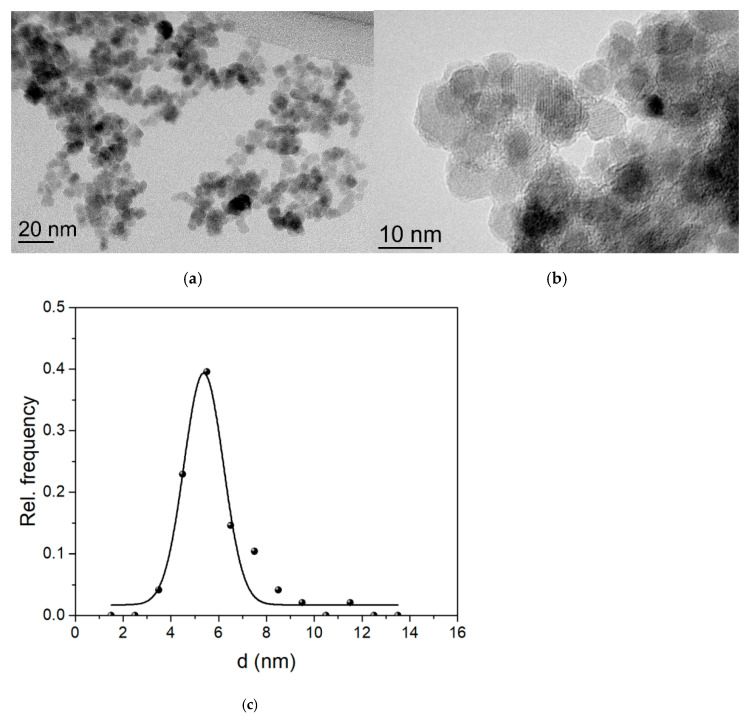
(**a**,**b**) TEM images of Fe_3_O_4_@CA MNPs; (**c**) corresponding NP size distribution from the TEM.

**Figure 6 nanomaterials-11-02965-f006:**
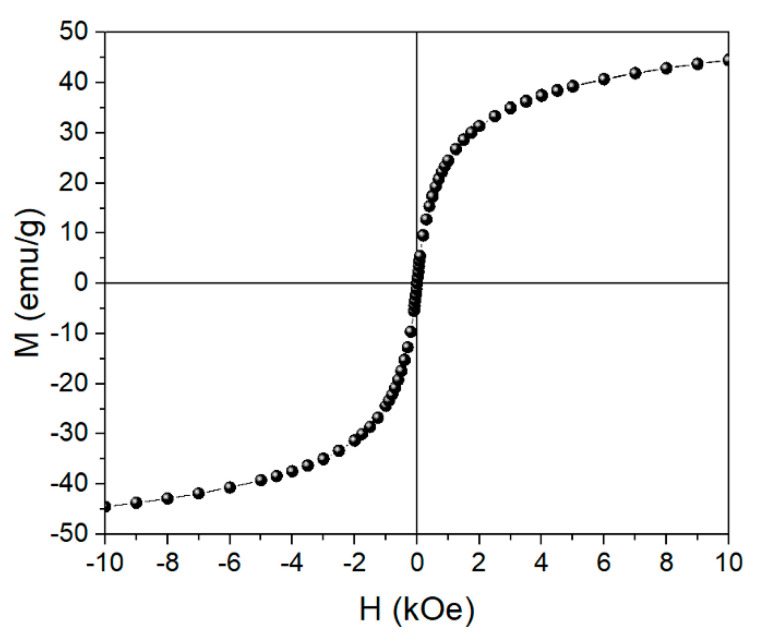
The room-temperature magnetization curve of the Fe_3_O_4_@CA MNPs (the line serves as a guide to the eye).

**Figure 7 nanomaterials-11-02965-f007:**
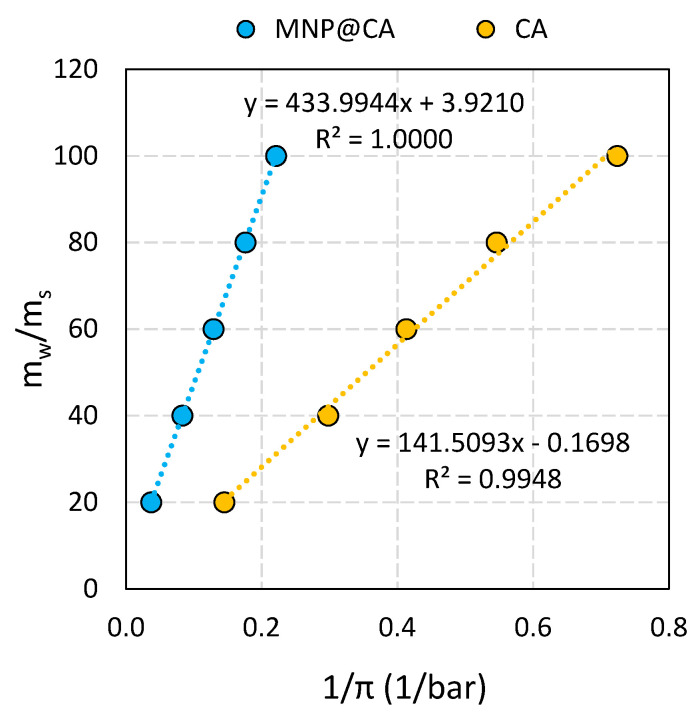
Non-ideality analysis of neat CA (orange) and MNP@CA nanocomposites (blue). *m*_W_ and *m*_S_: mass of water and mass of solute, π: osmotic pressure. Dashed lines represent fits to Equation (8) (r > 0.995) and fitting parameters are given in Table 4.

**Figure 8 nanomaterials-11-02965-f008:**
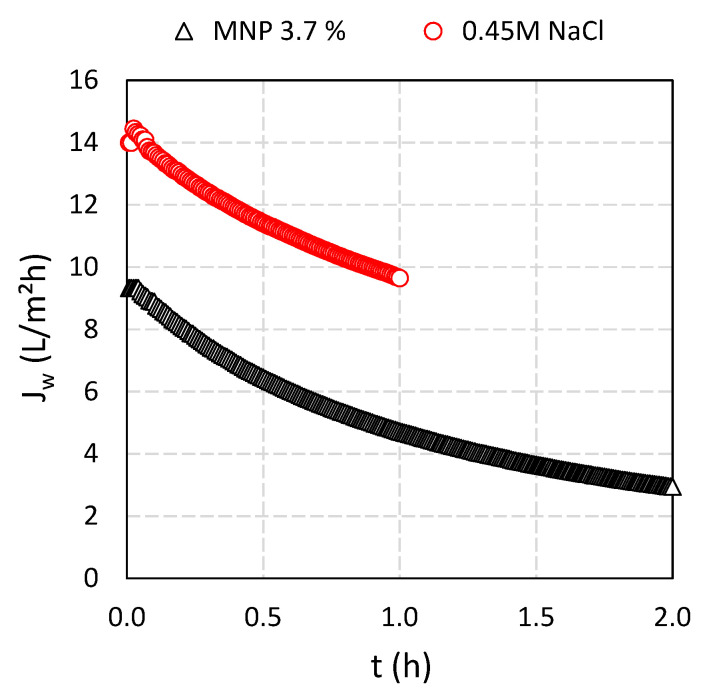
Water fluxes versus filtration time. Higher initial flux was reached using 0.45 M NaCl solution (14.1 LMH) when compared to Fe_3_O_4_@CA (9.3 LMH), respectively.

**Figure 9 nanomaterials-11-02965-f009:**
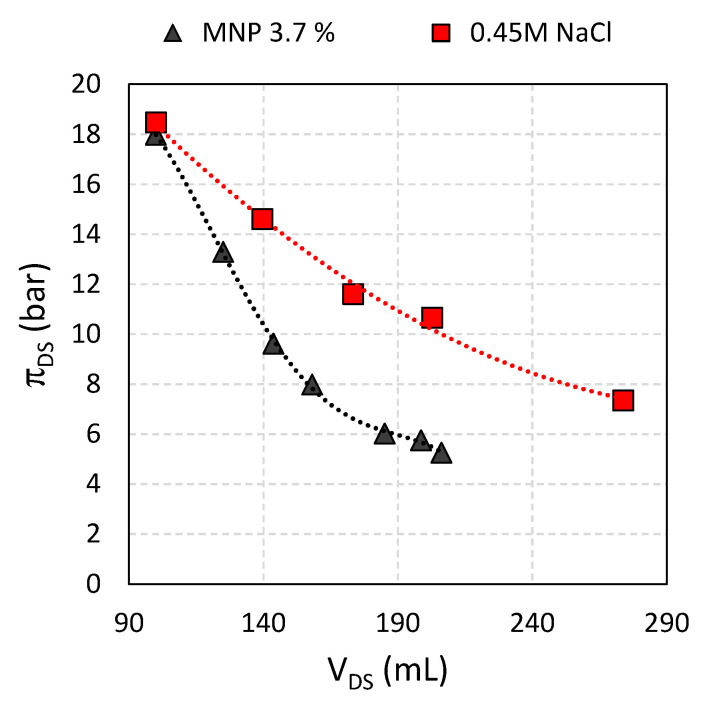
DS osmotic pressures with increased volume, for 0.45 M NaCl and Fe_3_O_4_@CA as DS and DI water as FS, respectively.

**Figure 10 nanomaterials-11-02965-f010:**
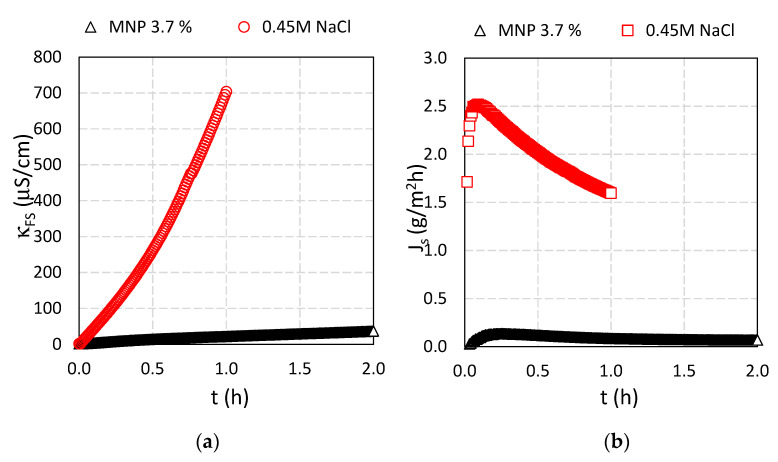
(**a**) Conductivity of FS and (**b**) revere salt fluxes during the time for 0.45 M NaCl and Fe_3_O_4_@CA as DS and DI water as FS, respectively.

**Figure 11 nanomaterials-11-02965-f011:**
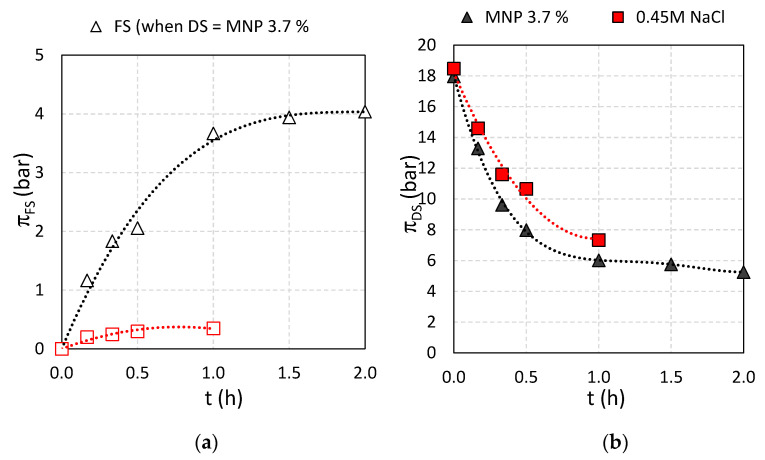
Osmotic pressures as function of time for (**a**) FS and (**b**) DS for 0.45 M NaCl and Fe_3_O_4_@CA as DS and DI water as FS, respectively.

**Figure 12 nanomaterials-11-02965-f012:**
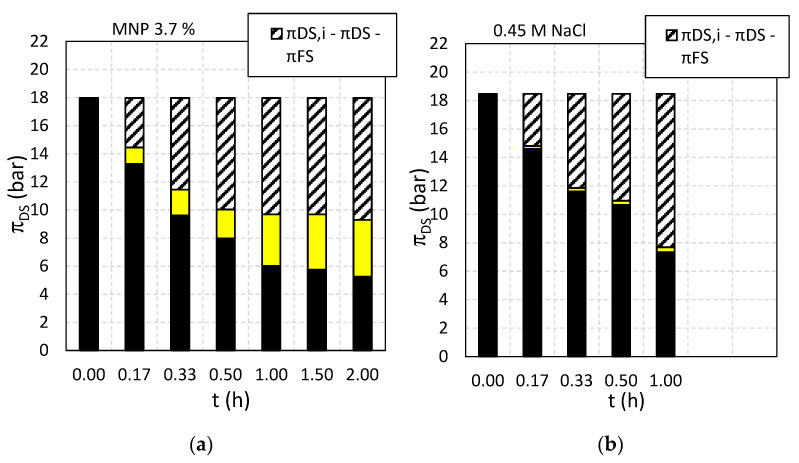
Osmotic pressures when DS is (**a**) Fe_3_O_4_@CA and (**b**) 0.45 M NaCl, during the time. Black columns present the osmotic pressure in the DS during filtration, which is decreasing because of two factors: (i) outcoming salts from DS to FS (yellow part of columns) and (ii) incoming water from the FS that is diluting DS (hatched part of columns).

**Figure 13 nanomaterials-11-02965-f013:**
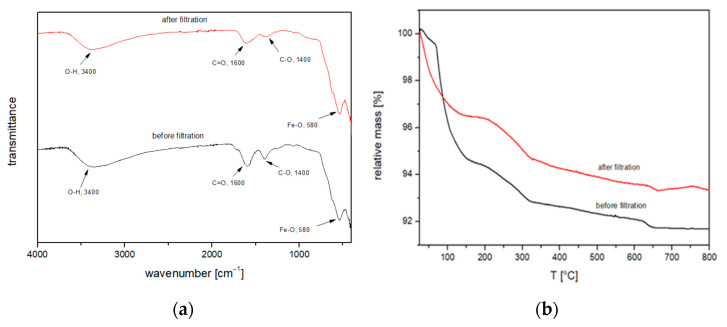
(**a**) FTIR spectrum of Fe_3_O_4_@CA MNPs before (red) and after (black) FO process and (**b**) TGA analysis curve of Fe_3_O_4_@CA after FO process.

**Table 1 nanomaterials-11-02965-t001:** Experimental overview.

	DS	FS	*V*_DS_ (mL)	*V*_FS_ (mL)
1	MNP (3.7% *w*/*w*)	DI water	100	250
2	0.45 M NaCl	DI water	100	250

**Table 2 nanomaterials-11-02965-t002:** Conductivity, pH and osmotic pressure values of Fe_3_O_4_@CA MNPs after synthesis and washing cycles.

	Conductivity (mS/cm)	pH	Osmotic Pressure (bar)
After synthesis	69.4	12.5	33.2
Drained water	70.8	12.9	32.2
1. Cleaning with DI water	28.8	12.4	10.3
Drained water	27.7	12.5	10.2
2. Cleaning with DI water	16.1	12.1	5.4
Drained water	15.5	12.2	5.4
3. Cleaning with DI water	7.6	11.9	2.5
Drained water	7.3	11.9	2.6
4. Cleaning with DI water	3.3	11.6	0.9
Drained water	2.7	11.4	0.9
5. Cleaning with ethanol	0.38	11.9	89.6
Drained ethanol	/	/	/
6. Cleaning with DI water	0.62	11.3	/
Drained water	0.5	11.3	/
Prepared DS	0.77	10.3	13.2

**Table 3 nanomaterials-11-02965-t003:** Concentration, pH, osmotic pressure, size, surface ligand concentration, IEP, and magnetic saturation of Fe_3_O_4_@CA MNP’s.

Sample Name	Concentration [%]	pH	Osmotic Pressure [bar]	*d*_x_ [nm]	*n*_S_ [molecules/nm^2^]	IEP	*M*_s_ [emu/g]
Fe_3_O_4_@CA	3.7	7	18.7	3–7	0.904	4.93	44

**Table 4 nanomaterials-11-02965-t004:** The fitting parameters of the non-ideality analysis Fe_3_O_4_@CA.

	*S*	*I*
MNP@CA	434	3.92
CA	142	−0.17

**Table 5 nanomaterials-11-02965-t005:** The initial and final volumes of the DS and FS.

	DS	FS	*V*_DS_ (mL) Initial	V_FS_ (mL) Initial	*V*_DS_ (mL)		*V*_FS_ (mL)	
					at 1 h	at 2 h	at 1 h	at 2 h
1	MNP	DI water	100	250	185.1	206.2	163.6	142.5
2	0.45 M NaCl	DI water	100	250	273.7	/	75.2	/

## Data Availability

Data available on request.

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
