# Peer review of "Superparamagnetic Fe3O4@CA Nanoparticles and Their Potential as Draw Solution Agents in Forward Osmosis"

_nanomaterials, 2021, doi:10.3390/nano11112965_

Round 1
Reviewer 1 Report
Authors synthesized Fe3O4@CA MNPs co-precipitation and characterized them, and investigated their application in forward osmosis. As a whole, the manuscript provided many scientific data, so it can be considered to publish after major revision.
- Abstract: Some abbreviations need full names.
- Many subscripts need be revised in all the manuscript.
- the symbola around the reference numbers should be revised.
- Some typefaces were wrong.
- Line 186-193, the relative content can be replaced into Materials and Methods section.
- The explanation for TGA curve was problematic, because Fe3O4 can be oxidized to Fe2O3 at 450 oC, so the estimated total number of COOH groups was problematic.
- Figure 4: the attribution of two curves was indistinct, authors should revise one of ordinate and the relative coordinate axis to red color, or label arrows for two curves.
- An XRD measurement was necessary for Fe3O4@CA MNPs.
- The osmosis examination data need be explained in the theoretical level.

Author Response
Authors synthesized Fe3O4@CA MNPs co-precipitation and characterized them, and investigated their application in forward osmosis. As a whole, the manuscript provided many scientific data, so it can be considered to publish after major revision.
- Abstract: Some abbreviations need full names.
In the draft we have checked the abbreviations mentioned and, where necessary, supplemented them with full names.
- Many subscripts need be revised in all the manuscript.
We apologize, but this shortcoming occurred during the conversion of the article to the template required by the journal Nanomaterials. Throughout the article, we have corrected the subscripts.
- the symbola around the reference numbers should be revised
The symbol has been correctly replaced with square brackets [] throughout the article.
- Some typefaces were wrong.
Throughout the article, we have unified the font type and size.
- Line 186-193, the relative content can be replaced into Materials and Methods section.
We would like to keep the content in lines 186-193 there since the content provide a 'roadmap' for the Results section.
- The explanation for TGA curve was problematic, because Fe3O4 can be oxidized to Fe2O3 at 450 oC, so the estimated total number of COOH groups was problematic.
Explanation:
Thank you for your comment. There was an error in calculation of estimated number of COOH groups, attached to MNPs. Accidentally we took into account the 5 % of water instead of additional 3 % due to the oxidation of magnetite to hematite.
The weight loss curve observed in a range of 100 to 800 °C is attributed to the evaporation of water, the decomposition of the bound citrate molecules on the Fe3O4 MNP’s surfaces 3 % (4, 5) and due to oxygen during magnetite oxidation 3,3 %. The initial weight loss below 160 °C due to the evaporation of physically adsorbed water was 5 % and the organic content of coated Fe3O4@CA was corrected to 6.3 %.
Also the number of molecules/nm2 (Table 3) was corrected to: 0.904.
- Figure 4: the attribution of two curves was indistinct, authors should revise one of ordinate and the relative coordinate axis to red color, or label arrows for two curves.
In Figure 4, we have changed the color of the ordinate indicating the Z-average to red and also used arrows to indicate which curve corresponds to the ordinate.
- An XRD measurement was necessary for Fe3O4@CA MNPs.
We did not use XRD to show that Fe3O4 nanoparticles were successfully coated with citric acid because the XRD's of bare nanoparticles and coated nanoparticles with CA are similar. We attach references to support this fact, see (6, 7).
- The osmosis examination data need be explained at the theoretical level.
We are not entirely sure what the reviwer means here. In section 3.4.1, below Fig. 9, the following paragraph was added to explain the 'osmosis examination data' at a theoretical level.
“The suitability of a draw solute is defined by its ability to develop osmotic pressure, which leads to higher water flux, and lower diffusion to the feed side. From the molecular view, the size and ionic structure of the draw solute defines its applicability in the FO process. The lower molecular size increases the diffusivity of the DS and reduces the internal concentration polarization that leads to the higher water flux, while the reverse salt flux is also enhanced. The larger molecules are designed as DS to prevent reverse salt diffusion.
In this study, small molecules such as NaCl that were used for comparison generated high osmotic pressure at low solution viscosities and mitigate ICP due to their high reverse solute flux [20]. “

Reviewer 2 Report
The manuscript “Superparamagnetic Fe3O4@CA nanoparticles and their potential as draw solution agents in forward osmosis” by Petrinic et al. represents a very interesting approach towards magnetic nanoparticles being used for osmotic driven processes.
I genuinely enjoyed reading this manuscript, which is original and described reproducibly. The references are balanced and the manuscript is written well.
However, there are a few issues with the manuscript as well.
There are a few typos (e.g. Fourier written small instead of captialized). The switch between fonts is really annoying and breaks the reading flow. Please use only one font throughot the manuscript.
The same goes for tables, figures and equations.
My major critique is that the novel results (section 3.4) are not at all discussed with literature.
I am sure that there is literature which describes similar effects as those observed in your forward osomosis study. Even if not called osmosis but mass transport. You should definitely discuss other literature.
Since you mention the stabilization of the nanoparticles, I am intrigued on how non-stabilized particles would affect the osmotic behavior? Is there a possibility that you can compare this behavior, or at least discuss it?
In the conclusion you can also give an outlook on how the observed phenomenon might be used in the future.
Author Response
The manuscript “Superparamagnetic Fe3O4@CA nanoparticles and their potential as draw solution agents in forward osmosis” by Petrinic et al. represents a very interesting approach towards magnetic nanoparticles being used for osmotic driven processes.
I genuinely enjoyed reading this manuscript, which is original and described reproducibly. The references are balanced and the manuscript is written well.
However, there are a few issues with the manuscript as well.
- There are a few typos (e.g. Fourier written small instead of captialized). The switch between fonts is really annoying and breaks the reading flow. Please use only one font throughot the manuscript. The same goes for tables, figures and equations.
Thank you very much for your comment. There were errors in transcribing the text into the template requested by the journal Nanomaterials. We have re-checked the text of the article and corrected errors related to font size, breaks, and spaces. The font has been fully unified, and we have also corrected tables, figures and equations.
- My major critique is that the novel results (section 3.4) are not at all discussed with literature. I am sure that there is literature which describes similar effects as those observed in your forward osomosis study. Even if not called osmosis but mass transport. You should definitely discuss other literature.
In the section 3.4. a new paragraph (below) was added which describes similar effects as those observed in our study. Two new references were added too (# 58 and # 59).
“When comparing with other studies where magnetic nanoparticles were used a DS in FO, several studies have focused on poly-sodium acrylate (PSA) coated MNPs as DS. Thus a water flux of 5.3 LMH and an DS osmotic pressure of 11.4 bar has been reported using PSA-MNPs at 0.078 %, wt % (~ 1.3 g/L) DS concentration [58]. Ge and co-workers [59] reported an osmotic pressure of 11–12 bar at a concentration of ~15 %, wt % PSA (Mn 1800) as DS. Similar osmotic pressures were obtained with 24–48 %, wt % PSA DS (in the form of free polyelectrolyte osmotic agent) [31]. The significant differ-ence in osmotic pressure between PSA-MNP solutions and free PSA solutions reflect the non-ideality of PSA-MNP solutions similar to what we observe in this study. However, the robustness of PSA-MNP based DSs (i.e. their ability to maintain their osmotic pressure after repeated concentrations) is still an issue [29].
In our previous study [29] a 7 % solution of robust PSA-coated MNP’s yielded an initial Jw of 4.2 LMH and an osmotic pressure of 9 bar and a reverse solute flux Js of 0. 05 GMH. Thus, the results presented here where Fe3O4@CA MNPs a 3.7% solution yielded approximately a two-fold higher initial Jw (Fig. 8) and osmotic pressure (Fig. 9) implies that the Fe3O4@CA MNPs DS is about four-fold more potent as DS compared to a DS based on PSA-coated MNPs. However, the nominal reverse solute flux for Fe3O4@CA of 0.08 GMH (Fig 10b) is slightly higher than for PSA-coated MNP based DS.”
- Since you mention the stabilization of the nanoparticles, I am intrigued on how non-stabilized particles would affect the osmotic behavior? Is there a possibility that you can compare this behavior, or at least discuss it?
As bare magnetite nanoparticles are highly susceptible to air oxidation and are easily aggregated in aqueous systems we pointed out in our previous work (1).
Aqueous dispersions of bare magnetite nanoparticles have negligible osmotic pressure, even at the maximum concentration attainable (~50 %, wt %) (2) and have an isoelectric point close to neutral pH, therefore, they are not stable and aggregate in aqueous solutions (3). Therefore, magnetic nanoparticles (MNPs) need to be functionalized with molecules that generate osmotic pressure and stabilize the dispersion.
- In the conclusion, you can also give an outlook on how the observed phenomenon might be used in the future.
A sentence was added in the Conclusion section on how the observed phenomenon (conductivity of the feed solution due to the same low level before and after FO filtration, and therefore negligible reverse solute flux of Fe3O4@CA MNPs) might be used in the future:
»Therefore, FO can be considered as a potential candidate for a broad range of concentration applications where current technologies still suffer from critical limitations. »
